# Effect of Polymeric Nanoparticles with Entrapped Fish Oil or Mupirocin on Skin Wound Healing Using a Porcine Model

**DOI:** 10.3390/ijms23147663

**Published:** 2022-07-11

**Authors:** Tomáš Komprda, Zbyšek Sládek, Monika Vícenová, Jana Simonová, Gabriela Franke, Břetislav Lipový, Milena Matejovičová, Katarína Kacvinská, Cristina Sabliov, Carlos E. Astete, Lenka Levá, Vendula Popelková, Andrej Bátik, Lucy Vojtová

**Affiliations:** 1Department of Food Technology, Mendel University in Brno, Zemedelska 1, 613 00 Brno, Czech Republic; jana.simonova@mendelu.cz (J.S.); gabriela.franke@mendelu.cz (G.F.); milena.matejovicova@mendelu.cz (M.M.); vendula.popelkova@mendelu.cz (V.P.); 2Department of Animal Morphology, Physiology and Genetics, Mendel University in Brno, Zemedelska 1, 613 00 Brno, Czech Republic; zbysek.sladek@mendelu.cz (Z.S.); andrej.batik@mendelu.cz (A.B.); 3Department of Infectious Diseases and Preventive Medicine, Veterinary Research Institute, Hudcova 296/70, 621 00 Brno, Czech Republic; vicenova@vri.cz (M.V.); leva@vri.cz (L.L.); 4Department of Burns and Plastic Surgery, Faculty of Medicine, Institution Shared with University Hospital Brno, Masaryk University, 625 00 Brno, Czech Republic; bretalipovy@gmail.com; 5Central European Institute of Technology, University of Technology, Purkynova 123, 612 00 Brno, Czech Republic; katarina.kacvinska@ceitec.vutbr.cz (K.K.); lucy.vojtova@ceitec.vutbr.cz (L.V.); 6Department of Biological and Agricultural Engineering, Louisiana State University, Baton Rouge, LA 70803, USA; csabliov@agcenter.isu.edu (C.S.); castete@agcenter.isu.edu (C.E.A.)

**Keywords:** cutaneous wounds, polyunsaturated fatty acids n-3, nanoparticles, poly(lactic-co-glycolic) acid, mupirocin, collagen, hydroxyproline, cyclooxygenase-2, transforming growth factor

## Abstract

The utilization of poly(lactic-co-glycolic) acid (PLGA) nanoparticles (NPs) with entrapped fish oil (FO) loaded in collagen-based scaffolds for cutaneous wound healing using a porcine model is unique for the present study. Full-depth cutaneous excisions (5 × 5 cm) on the pig dorsa were treated with pure collagen scaffold (control, C), empty PLGA NPs (NP), FO, mupirocin (MUP), PLGA NPs with entrapped FO (NP/FO) and PLGA NPs with entrapped MUP (NP/MUP). The following markers were evaluated on days 0, 3, 7, 14 and 21 post-excision: collagen, hydroxyproline (HP), angiogenesis and expressions of the COX2, EGF, COL1A1, COL1A3, TGFB1, VEGFA, CCL5 and CCR5 genes. The hypothesis that NP/FO treatment is superior to FO alone and that it is comparable to NP/MUP was tested. NP/FO treatment increased HP in comparison with both FO alone and NP/MUP (day 14) but decreased (*p* < 0.05) angiogenesis in comparison with FO alone (day 3). NP/FO increased (*p* < 0.05) the expression of the CCR5 gene (day 3) and tended (*p* > 0.05) to increase the expressions of the EGF (day 7, day 14), TGFB1 (day 21) and CCL5 (day 7, day 21) genes as compared with NP/MUP. NP/FO can be suggested as a suitable alternative to NP/MUP in cutaneous wound treatment.

## 1. Introduction

Wound healing is a complex programmed sequence of cellular and molecular processes consisting of inflammation, cell migration, angiogenesis, synthesis of provisional matrix, collagen deposition and re-epithelialization [1]. It proceeds in four overlapping phases: inflammation, coagulation, tissue formation and tissue remodeling [2]. The success rate of the healing process is usually evaluated based on the following markers: the extent of collagenesis (semi-quantified using the Sirius red staining [3] content of hydroxyproline, a biochemical marker for collagen and an indicator of the progression of healing [2]); the extent of angiogenesis (usually evaluated by the semi-quantitative immunohistochemistry procedure detecting alpha smooth muscle actin, α-SMA, in the blood vessel wall [4]); and the expressions of genes coding for factors that are active during the process of wound healing, such as vascular endothelial growth factor (VEGF), epidermal growth factor (EGF), transforming growth factor beta (TGFB [5]), cyclooxygenase 2 (COX2 [4]), collagen type I alpha 1 (COL1A1), collagen type III alpha 1 (COL3A1 [6]), and CC-chemokines and their receptors (CCL5, CCR5 [7,8]).

Rodents are often used as an animal model for testing drugs active in wound healing in humans [9,10]. However, the driving force behind wound closure in rodents is contraction [11], whereas the closure of human wounds is primarily accomplished through the proliferation and migration of cells at the wound edge. The wound-healing process in pigs is similar to that of humans, making porcine models more suitable for the study of treatments that need to be evaluated for eventual use in humans [12].

Among many types of polymeric materials, poly(lactic-co-glycolic) acid (PLGA) was introduced as a promising carrier of wound-healing agents [13]. PLGA is suitable for the production of nanoparticles (NPs) for clinical applications, because it protects loaded drugs from biological degradation, and due to an enhanced stability and sustained release, it allows a reduction in the administrated dose [14]. The excellent biocompatibility and adjustable mechanical properties of PLGA have resulted in its wide use as a drug carrier that improves drug delivery and absorption by oral or parenteral administration [15].

Among other treatments, antibiotics or/and anti-inflammatory drugs can be incorporated into NPs to improve and accelerate the wound-healing process. Mupirocin is a commonly used antibiotic that is effective against bacterial skin infections and promotes wound healing [16]. Budhiraja et al. [17] used mupirocin-loaded chitosan microspheres to accelerate the wound-healing activity of a collagen scaffold. Muciprocin was also loaded in microspheres prepared using polyvinyl alcohol (PVA) and sodium alginate by Solak et al. [18], who quantified its release and antibacterial properties against *Staphylococcus aureus*.

As far as anti-inflammatory substances potentially usable as components of the NP-based drug-delivery carriers are concerned, an interesting possibility is fish oil (FO), due to its anti-inflammatory and anti-oxidative properties [19]. FO entrapment in NPs is still rather uncommon, but it was recently used by Rakotoarisoa et al. [20] in curcumin- and FO-loaded spongosome and cubosome NPs. The anti-inflammatory effects of the active components of FO, n-3 long-chain polyunsaturated fatty acids (LC-PUFA), on wound healing (recently reviewed by [21]) are a result, at least in part, of the competition with arachidonic acid in eicosanoid synthesis [22] and of the modulation of signaling pathways mediated by transcription factors PPARα, PPARγ and NF-κB [23].

Despite the fact that the effects on wound healing of PLGA NPs and FO alone, respectively, have been repeatedly tested, the effect of PLGA with entrapped FO (PLGA-FO) NPs on an animal model of cutaneous wound healing has not been previously investigated. The objective of this study was to test the following hypotheses using a porcine model: FO entrapped in PLGA NPs improves selected markers of cutaneous wound healing in comparison with FO alone; PLGA/FO NPs demonstrate effects comparable to those of PLGA/MUP NPs.

## 2. Results

### 2.1. Collagenous Tissue Maturation

Collagenous tissue was semi-quantified by Sirius red staining, and its maturation was assessed by the stain intensity. In Figure 1, collagen type III and collagen type I are characterized by the light-red color and the dark-red color, respectively. It is evident that most of the collagen type I was detected on days 14 and 21.

Neither NPs with fish oil nor FO alone had a significant effect on the amount of newly formed collagen in any of the tested time intervals. The effects of all experimental interventions either did not differ from the control (dressing alone) or resulted in a lesser amount of collagen than that in the control (Table 1).

Samples were classified in the following three grades of the red intensity in comparison with the red intensity of the control sample: “0” (equal intensity—same maturation); “−” (lower intensity—less advanced maturation); “+” (higher intensity—more advanced maturation; this grade was not detected in any of the experimental treatments).

The concentration of hydroxyproline (HP; a major collagen component) in the healing tissue did not show a significant increase for any of the experimental treatments in comparison with the control for any of the time intervals tested (Figure 2). However, NPs with entrapped FO significantly increased (*p* < 0.05) HP concentration in the healing tissue 14 days post-excision in comparison with both free FO and NP/MUP treatments (Figure 2).

### 2.2. Angiogenesis

The extent of new blood vessel formation in the healing skin tissue was semi-quantitatively evaluated by immunohistochemical (IHC) labeling for alpha-smooth muscle actin (Figure 3). Representative IHC sections of the healing tissue are shown in Figure 4.

NPs with entrapped FO had lower (*p* < 0.05) α-SMA expression than the controls at the end of the observation period (21st day). On the other hand, on days 3 and 14, FO treatment had a greater (*p* < 0.05) number of new vessels relative to the control. The number of new vessels in the samples treated with NP/FO was lower (*p* < 0.05) than for FO alone both at the beginning (day 3) and at the end (day 21) of the experiment. NP/FO treatment also decreased (*p* < 0.05) the number of new vessels as compared with NP/MUP on day 21.

It is interesting that, when the number of newly formed blood vessels and hydroxyproline content, respectively, were evaluated irrespective of the chosen treatment, the courses of both traits during the whole time period tested were similar, which was demonstrated by the highly significant correlation coefficient of HP vs. α-SMA: r = 0.48 (*p* < 0.01).

### 2.3. Gene Expression

No differences between NP/FO and free FO (*p* > 0.05) were observed in the expressions of the *COX2*, *COL1A*1, *TGFB1* and *CCR5* genes, for any of the time intervals tested. On the other hand, at the early stage of healing (day 3), free FO increased (*p* < 0.05) the expressions of the *EGF*, *VEGFA* and *COL*3*A1* genes in comparison with NP/FO.

As far as the comparison of NP/FO with NP/MUP is concerned, entrapped FO decreased (*p* < 0.05) the expression of the *EGF* gene by day 3 and tended to decrease (*p* > 0.05) the expression of the *VEGFA* gene by day 21. On the other hand, the tendency (*p* > 0.05) of entrapped FO to increase gene expression as compared with NP/MUP is suggested in the case of *EGF* (day 7, day 14), *TGFB1* (day 21) and *CCL5* (day 7, day 21). The only other significant differences between NP/FO and NP/MUP were found in the expression of the *CCR5* gene; entrapped FO increased (*p* < 0.05) its expression in the early stage (day 3), and *CCR5* expression tended to be higher in the NP/FO samples (*p* > 0.05) at the end of the observation period (Figure 5).

### 2.4. Microbiological Analysis

When summed over all treatments, the total microbial counts (TMCs) increased (*p* < 0.05) from 0 log CFU/25 cm^2^ (day 0; samples were sterile immediately after excisions) to 1.14 (day 7), 2.25 (day 14) and 4.11 log CFU/25 cm^2^ 21 days after excision (Figure 6). Most of the differences among treatments were insignificant for all tested time intervals (*p* > 0.05), with the exception of FO, which had a greater (*p* < 0.05) TMC on day 7 than all other treatments.

The counts of Streptococcus pyogenes tended to increase (*p* = 0.09) from 0 log CFU/25 cm^2^ (day 0) to 0.20 (day 7) and to 0.47 (day 14) and then significantly increased (*p* < 0.05) to 3.24 log CFU/25 cm^2^ at the end of observation. Similarly to the TMC, the only treatment that was significantly different from the others was FO, which had higher Streptococcus pyogenes counts 14 days after excision than the other treatments (*p* < 0.05).

In the case of *Staphylococcus aureus*, though the differences among treatments did not reach statistical significance, 14 days after excision, the bacterial counts for all experimental treatments tended to be lower than (from *p* = 0.07 to *p* = 0.09) the counts for the control. At the end of observation (day 21), both FO and NP/FO had bacterial counts that were nearly significantly greater (*p* = 0.06) counts of S. aureus than counts for MUP and NP/MUP.

No differences among treatments (*p* > 0.05) were found in the counts of *E. coli* in any of the tested time intervals; *E. coli* was not detected in the C, NP, FO and MUP samples (results not shown).

## 3. Discussion

Generally speaking, the hypothesis, tested in the present study, that a combination of PLGA NPs with entrapped fish oil would improve the markers of the healing of cutaneous excision wounds more than PLGA NPs alone or FO alone was not supported. This is contrary to data obtained by Chereddy et al. [24], who applied PLGA/curcumin NPs in a full-thickness excisional wound-healing mouse model and found twofold higher wound-healing activity than with PLGA alone or curcumin alone. They observed higher re-epithelialization, granulation-tissue formation and anti-inflammatory potential for wounds treated with PLGA/curcumin NPs.

In the present study, the relative lack of significant differences among the tested treatments or among the treatments and the control can be considered from at least three viewpoints: animal model, control empty collagen scaffolds and amounts of the active substances applied. Most experiments testing the effects of different active substances on excisional wound healing are performed on rodents [13,24,25,26,27], using a very homogeneous set of inbred animals. Because the wound-healing process in humans and pigs is similar (and different from the healing process in rodents), we used a porcine model to evaluate the potential efficacy of the tested treatments in humans [12]. However, though the set of pigs used in the present experiment was as homogeneous as possible (same producer, single sex, same age and similar weight), the differences among animals often decreased the significance of the differences among treatments.

The dressings that we used in our experiments may have affected our results. In a study of wound healing in rats, Liu et al. [25] found that open wounds treated with PLGA/collagen nanofiber dressings healed more quickly than wounds treated with commercial dressings. In our case, collagen scaffolds, specifically prepared for the present experiment at the Central European Institute of Technology in Brno, greatly improved healing when used alone, and that overall improvement may have partly obscured the independent effects of the active substances.

The rather inconclusive results regarding NP/FO combination (or FO alone) are not too surprising; according to Bradberry et al. [28], the effects of the active components in FO (LC-PUFA n-3; eicosapentaenoic acid (EPA) and docosahexaenoic acid (DHA)) on wound healing are not clear yet. LC-PUFA n-3 was found to inhibit skin wound healing in one study [29] but was found to have no effect in another [30]. Elevated levels of DHA slowed inflammation resolution and impaired the quality of healed skin tissue in a mouse model [26]. Using a similar (mouse) model, skin lesions topically treated with alpha-linolenic acid (n-3) were significantly larger at 120 h after injury in comparison with those treated with oleic acid (n-9). On the other hand, Arantes et al. [31] reported that topically applied DHA improved wound healing (with the potential activation of GPR120 in the process).

As far as the concentrations of the active substances in the scaffolds used in the present study are concerned, based on the results of the preliminary experiment using three pigs (data not shown), the amounts of the active substances were increased; the values are presented in Section 4.4 (see below). However, more conspicuous differences among treatments could not be ruled out, if the concentrations were increased even more.

### 3.1. Collagenous Tissue Maturation

Collagens are present in the dermis as fibrillar proteins that have enormous tensile strength [32]. Collagen deposition is a fundamental step in wound healing that provides the matrix for angiogenesis and tissue remodeling. In normal skin, collagen fibrils are composed of both collagen I and III, with collagen III comprising ~20% of the total [33]. We detected the predominance of collagen III type in the early stages of wound healing 3 and 7 days after excision in all treatments, including control samples. This is consistent with the known fact that myofibroblasts lay down collagen III during the early stages of granulation-tissue formation. During the initial phase of wound healing, collagen III expression increases more than the collagen I expression, resulting in an increased ratio between the two collagen subtypes from 20% up to 50% of collagen III [34]. Wang et al. [35] reported significant increases in collagen I mRNA levels on day 14 after excision, so this time interval appears to be significant in the dynamics of these changes. As our results show, during the maturation of the scar, the amount of collagen I steadily increased and reached the peak on the 14th and 21st days after excision.

In spite of these general trends in collagen expression, we did not find differences in collagen content among treatments. In addition, we did not observe a positive effect of PLGA on collagen deposition, as was found by Chereddy et al. [24] in a mouse study in which wounds were treated with PLGA–curcumin NPs. Differences among the animal models and drug loads used in the two studies could explain the contrary results. Similarly, the findings obtained by Gercek et al. [1] indicating that fish oil accelerates collagen formation were not confirmed in the present study; the same was true regarding the results of other experiments evaluating effects of nanoparticles, fish oil and antibiotics, respectively, on collagen deposition in wounds [36,37,38].

As far as collagen deposition is concerned, it is evident from the results of the present study that wound healing occurred physiologically in all treatments without any tendency for hypertrophic scarring or keloid formation. However, it should be noted that the complete process of skin-excision wound healing lasts much longer than the three weeks recorded in the present study; as many as six months may elapse before collagen content in the repaired tissue resembles normal skin [39].

### 3.2. Angiogenesis

The restoration of the vascular system of the skin is a complex cascade of cellular, humoral and molecular events in the wound bed to restore the nutritive perfusion. Therefore, angiogenesis is a crucial step that allows nutrients to be delivered to granulation-tissue components [9]. In this study, we detected a small number of blood vessels in the control sample on the third day after excision. This was expected, as the proliferative phase of healing begins at this time. The number of vessels generally increased over time; the peak of vascularization was recorded on the 21st day in the control, NP alone and NP/MUP treatments, while the peak vascularization for FO, MUP and NP/FO treatments was observed on the 14th day.

In the present study, FO alone tended to elicit a higher level of angiogenesis than the other treatments across the observation period. This is consistent with the improved neovascularization observed in mice fed a fish-oil-enriched diet in an experiment conducted by Turgeon et al. [40]. Similarly, Mahmoud et al. [41] found out that a mixture of fish oil and honey enhanced epithelialization and neovascularization in wound healing in horses and donkeys. Shingel et al. [42] reported that a solid emulsion gel containing fish oil stimulated early angiogenesis and promoted wound repair in vivo. In the present study, FO treatment was associated with the significant (*p* < 0.05) increase in angiogenesis noted on the 14th day in comparison with day 7. In an experiment with rats that used the same temporal sampling as in our study, Tanideh et al. [38] observed more granulation tissue, a higher level of epithelialization, a greater number of blood vessels, more fibroblasts and more collagenous fibers in the group treated with fish oil on the 14th day than on the 7th day.

A similar trend was also observed with MUP treatments, with the peak of the number of vessels being observed on the 14th day. This is not surprising, because MUP also promotes angiogenesis during wound healing [43,44] and increases VEGF, which is a key mediator in angiogenesis [45]. However, a higher number of vessels was detected for MUP alone than in NP/MUP on the 14th day. This is not consistent with the data obtained by Golmohammadi et al. [46] for the MUP complexes with inorganic (Se) nanoparticles, which were found to have a positive effect on angiogenesis.

The number of vessels on day 14 was twice as high in the NP/FO samples compared with the NP samples (though the difference was not significant; *p* > 0.05) in our study. This is interesting, because nanoparticles themselves, both PLGA and metal NPs, support angiogenesis [47,48]. PLGA NPs supply lactate, whose increased levels in wounds stimulate angiogenesis, and is an important signal for collagen synthesis [13]. Several research groups have used PLGA nanoparticles to load pro-angiogenic biomolecules to improve angiogenesis and to accelerate tissue healing (for details, see [49]).

It is important to mention that the present experiment analyzed angiogenesis by counting the number of blood vessels, but did not evaluate their functionality. In fact, many blood vessels formed during wound healing are not perfused [50], so the decreased blood vessel density by the selective elimination of non-perfused blood vessels should not have a significant effect on wound healing.

### 3.3. Gene Expression

The expressions of the genes coding for the following factors relevant to wound healing were quantified: COX2, EGF, COL1A1, COL3A1, TGF-ß1, VEGFα, CCL5 and CCR5.

COX2 is a critical enzyme involved in the inflammatory response to wound injury [51]. Elevated DHA content either in transgenic mice capable of producing endogenous PUFA n-3 or in wild-type mice orally supplemented with DHA-enriched fish oil slowed inflammation resolution [26]. However, neither FO alone nor NP/FO showed any significant effect on the expression of the COX2 gene in the present study. The inhibition of COX2 activity (using transgenic mice) delayed re-epithelialization and angiogenesis in an experiment by Pan et al. [51]. Cardoso et al. [27] reported increased COX2 mRNA in the skin wounds of mice topically treated with PUFA n-3 but decreased COX2 expression in mice treated with monounsaturated oleic acid with a consequence of less pro-inflammatory lipid mediators being produced at the wound sites.

Epidermal growth factor (EGF) facilitates epidermal cell regeneration and plays an essential role in dermal wound healing through the stimulation of the proliferation and migration of keratinocytes and of fibroblast motility, the promotion of the formation of granulation tissue and the stimulation of collagenase activity [52]. Regarding collagenase, significant correlations (EGF × COL1A1 and EGF × COL3A1) were found in the present study (r = 0.27 and r = 0.42, respectively; *p* < 0.01). Both FO alone and NP/FO increased the expression of the EGF gene in comparison with MUP (Figure 5). The expressions of the COL1A1 and COL3A1 genes during wound repair in wild-type mice increased with recovery time and peaked on day 10 in an experiment by Pan et al. [51]. This course was different in the present experiment in pigs, where after the increase from the 3rd day to the 7th day, the expressions of both genes significantly decreased by day 14 and then again increased by day 21.

As far as the TGFB1 gene is concerned, we found a tendency of FO to show a higher expression 14 days post-excision than the control, but a similar effect of NP/FO was not observed (Figure 5). TGF-ß1 is considered a direct inducer of the transition of fibroblasts to myofibroblasts [53]. Its increased expression during early wound healing (day 4) in mice increased angiogenesis [7]; in this time interval, angiogenesis is critical, but it is pathological if it persists in later stages [7]. The over-expression of this cytokine may be associated with the protraction of the wound-healing process [54]. However, in the present study (when evaluated irrespective of the chosen treatment), the expression of the TGFB1 gene was low on day 3 and tended to be higher on days 7 and 21, and no pathologies were observed. On the other hand, taking into account that TGF-ß1 also stimulates collagen formation and the remodeling of the extracellular matrix [55], this increase in TGFB1 gene expression in the latest phases of wound healing in the present study can be considered positive. Similarly, Rezaii et al. [56], using a full-thickness punch wound model in rats, reported earlier wound closure, an increased formation of granulation tissue, increased neo-vascularization, greater collagen content and earlier re-epithelialization based on TGF-ß1 mRNA up-regulation on day 3 (early stage) as well as day 15 (late stage).

Another signal protein that stimulates the formation of blood vessels and has an important role in angiogenesis is VEGF; its expression in normal skin is absent, but cutaneous damage prompts a sharp up-regulation of VEGF expression [54]. In a study of rat full-thickness skin wound healing, Khalaf et al. observed the up-regulation of VEGF mRNA at the early stages of healing with a peak on day 7 and its sharp down-regulation on day 14. A different timing was demonstrated in the present study using a porcine model, e.g., NP/FO treatment increased VEGFA gene expression from day 3 to day 7, and its down-regulation was observed as late as from day 14 to day 21. Pan et al. [51] reported in a mice model of punch biopsy a peak of VEGFA gene expression on day 3.

Pro-angiogenic markers TGF-ß and VEGF are modulated by an inhibition of C-C chemokines [7], and the involvement of the chemokine system is crucial to skin wound healing [57]. TGF-ß expression was significantly higher in a group of mice treated with C-C chemokine inhibitor [7]. This contrasts with the results of the present study, where a positive correlation between the expressions of the genes coding for CCL5 and TGFB1 (r = 0.66; *p* < 0.01) and those coding for CCL5 and VEGFA (r = 0.53; *p* < 0.01) was found.

As far as the corresponding chemokine receptor is concerned, CCR5 deficiency impaired collagen production and neovascularization in a mouse model of skin wound, and mice lacking CCR5 showed reduced expressions of VEGF and TGF-ß [8], which is in agreement with the significant correlations of CCR5 × TGFB1 (r = 0.83) and CCR5 × VEGFA (r = 0.48) established in the present experiment. On the other hand, the expression of CCR5 increased by day 3 after injury, remaining at this level 6 days in the mouse experiment [8], but for FO and NP/FO treatments in the present study, we observed a relatively high CCR5 expression on day 3, which decreased by day 7 and remained low until day 14 (Figure 5).

### 3.4. Microbiological Markers

The antimicrobial effects of the biologically active substances present in fish oil (n-3 PUFA: EPA, DHA) are presumably due to the disruption of intercellular communication, the interruption of ATP production or the modification of the membrane properties of bacteria [58]. Moreover, EPA and DHA play significant roles in the down-regulation of the expression of the bacterial genes associated with biofilm formation [59]. However, these effects did not manifest themselves in the present experiment; FO treatment even tended to be associated with higher TMC and counts of *Streptococcus pyogenes* 7 and 14 days after excision, respectively (Figure 6).

We did not observe synergistic effects of either NP/FO or NP/MUP, contrary to the results obtained by [60], who reported synergistic antibacterial (against *Staphylococcus aureus* and *E. coli*) and wound-healing properties of harmala-alkaloid-rich fractions loaded into PLGA NPs coated with chitosan.

One factor possibly reducing the antimicrobial effects of the nanoparticles used in the present study was their negative charge; positively charged clindamycin-loaded PLGA-polyethylenimine NPs, in comparison with negatively charged NPs, adhered better to bacteria in the treatment of MRSA-infected wounds and accelerated the healing and re-epithelialization of wounds in a mouse model [61].

## 4. Methods and Materials

### 4.1. Animals

The study was conducted using ten female pigs of hybrid Large White (50%) × Landrace (50%; Bioprodukt Knapovec a.s., Ústí nad Orlicí, Czech Republic) aged 10 weeks and with a mean live weight of 45 kg. The experiment was performed in three consecutive phases using three, three and four pigs, respectively. The pigs were housed in an experimental stable in floored indoor pens of 290 cm × 343 cm (the height of the room was 280 cm) containing three animals for phases one and two; two stables containing two pigs each were used for phase three. The experiment was conducted in compliance with Czech National Council Act No. 246/1992 Coll. To prevent animal cruelty and with Amended Act No. 162/1993 Coll. and was approved by the Commission to protect Animals against Cruelty of Mendel University in Brno and the Ministry of Agriculture of the Czech Republic under statement No. 35537/2020-MZE-18134, serial No. Mze 2165.

The pigs were fed (one week prior to the start of the study and then during the experiment) a standard commercial feed mixture for pig fattening (De Heus, Marefy, Czech Republic) ad libitum twice daily and had free access to drinking water.

### 4.2. Experimental Design

Six variants of the treatment of skin-excision wounds were used: pure collagen scaffold (control, C); collagen scaffold enriched with empty poly(lactic acid-co-glycolic acid) (PLGA)-poly(vinyl alcohol) (PVA) nanoparticles (NPs) (NP); collagen scaffold with fish oil alone (FO); collagen scaffold with antibiotic mupirocin alone (MUP); collagen scaffold with PLGA-PVA NPs with entrapped fish oil (NP/FO); and collagen scaffold with PLGA-PVA NPs with entrapped mupirocin (NP/MUP). For our study, all FO treatments used commercial cod liver oil (*oleum jecoris aselli*) characterized by fatty acid content as % of the sum of total fatty acids as follows: 14:0 4.6, 16:0 11.1, 17:0 1.0, 18:0 2.9, 16:1 10.4, 18:1 19.5, 20:1 15.8, 18:2n-6 3.3, 18:3n-6 0.8, 20:2n-6 1.0, 20:4n-6 1.0, 18:3n-3 1.6, 20:5n-3 10.4, 22:5n-3 1.9 and 22:6n-3 13.6. MUP treatments used Sigma-Aldrich mupirocin (>92% powder; St. Louis, MO, USA).

### 4.3. Production of Nanoparticles

NPs were synthesized by emulsion evaporation. In the case of NPs with entrapped FO (NP/FO), the organic phase was prepared by mixing 440 mg of PLGA (*M*w 38,000–54,000 g/mol; lactide/glycolide ratio of 50:50; Sigma-Aldrich), 10 mL of dichloromethane (anhydrous: 99.8%; Sigma-Aldrich) and 200 mg of FO. Regarding NP/MUP, the organic phase was prepared by mixing 220 mg of PLGA, 5 mL of ethyl acetate and 11 mg of MUP. The aqueous phase of 2% PVA (98–99%; hydrolyzed; *M*w 31,000–50,000 g/mol; Sigma-Aldrich) was prepared by stirring (400 rpm) at 60 °C. The organic phase was drop-wise added to the aqueous phase by stirring at 220 rpm, and the formed emulsion was microfluidized four times at 30,000 psi using an M-110P microfluidizer (Microfluidics, Westwood, MA, USA). The solvent was then evaporated under vacuum using a rotary evaporator (Rotavapor R-124; Buchi Inc., New Castle, DE, USA) for 45 min (800 mm methylene chloride). NPs were then dialyzed for two days (25 °C) to remove excess PVA using a 300 kDa Spectra/POR CE membrane (Spectrum Rancho, Rancho Cucamonga, CA, USA). Consequently, the samples were mixed (1:1 *w*/*w*) with 100 mg of trehalose (dihydrate; 99.0%; *M*w 378.33 g/mol; Sigma-Aldrich), frozen for 30 min and then freeze-dried for two days (FreeZone Plus 2.5; Labconco Corporation, Kansas City, MO, USA). The NP powder was refrigerated (under −20 °C) and shielded from light until used. Empty NPs were prepared with the same procedure as in the case of NP/FO, only without fish oil. The content of FO in NP/FO particles was 50 mg/g, and that of MUP in NP/MUP was 7.5 mg/g.

The average sizes of NP, NP/FO and NP/MUP were 190.7, 233.3 and 141.1 nm; zeta potentials were −4.95, −68.1 and −21.9 mV; and polydispersity indices were 0.218, 0.351 and 0.336 A.U., respectively. Additional characteristics of the loaded nanoparticles are available in previous studies of NP/FO [62] and NP/MUP [63].

### 4.4. Dressings

Collagen (bovine type I; 8% aqueous gel; Collado, s.r.o., Brno, Czech Republic) was freeze-dried to obtain 100% collagen foam. Collagen scaffolds were prepared according to our previous work [64] using a slightly modified freeze-drying method, using 0.5 wt% collagen aqueous suspensions (ultrapure water—type II ISO 3696—prepared using an Elix 5 UV Water Purification System; Merck s. s r. o., Prague, Czech Republic). The required amount of the particular bioactive agent for each of the 5 × 5 cm dressings (FO alone, 5 mg; MUP alone, 23.4 μg; NP/FO, 50 mg; NP/MUP, 10 mg; empty NP, 25 mg) was slowly added to cold (4 °C) aqueous collagen suspensions. For the NP treatments, these amounts corresponded to 2.5 mg of FO entrapped in 50 mg of NP/FO and 75 μg of MUP in 10 mg of NP/MUP. Subsequently, the mixtures were homogenized using an IKA disintegrator (Ultra-Turrax T 18; IKA-Werke GmbH, Staufen, Germany), poured in 5 × 5 cm blister molds and freeze-dried in a Martin Christ Epsilon 2–10D lyophilizer (Osterode am Harz, Germany) at −35 °C under 1 mBar pressure for 15 h, followed by a secondary drying process at 25 °C under 0.01 mBar until Δp decreased (the change in pressure was up to 10%). Prior to in vivo testing, all foamed samples were sterilized with ethylene oxide.

### 4.5. Excisions and Sample Collection

Pigs were acclimatized for seven days. All surgical procedures were performed under general anesthesia. Premedication and analgesia during the surgical procedure were provided with the subcutaneous administration of butorphanol (0.1 mg/kg BW). Anesthesia was induced with a combination of tiletamine/zolazepam (2 mg/kg BW), ketamine (2 mg/kg BW) and xylazine (2 mg/kg BW) administered intramuscularly. General anesthesia was then maintained by intravenous propofol administration (8–15 mg/kg BW). Immediately after surgery, the second analgesic drug (meloxicam, 0.1 mg/kg BW) was administered subcutaneously and then continued at a dose of 0.1 mg/kg once a day for three consecutive days. Six full-thickness (epidermis, dermis) 5 × 5 cm cutaneous excisions on the dorsum in two para-sagittal planes were performed (Figure 7), and particular dressings were immediately applied. The positions of the six types of treatment (C, NP, FO, MUP, NP/FO and NP/MUP) on the dorsa of the ten pigs were randomized using GraphPad software [65]. The dressings were changed 3, 7 and 14 days post-excision, and the healing-tissue samples for analyses were collected on days 0 (only for microbiological analysis), 3, 7, 14 and 21 after excision. Samples for performing the following analyses were taken: collagenous tissue maturation, the concentration of hydroxyproline, the extent of angiogenesis, the expressions of eight genes coding for proteins relevant to skin tissue healing and the counts of four groups of microorganisms. Samples for histological (quantification of collagen fibers) and immunohistochemical (extent of angiogenesis) analyses were prepared as paraffin-embedded and cryosections, respectively.

### 4.6. Analysis of Total Collagenous Tissue Maturation

Tissue samples of healing skin were fixed in 10% buffered formalin, dehydrated by a gradual alcohol series, cleared in xylene, embedded in paraffin blocks and sectioned at 8 µm using a Leica RM2145 rotary microtome (Leica Microsystems, Wetzlar, Germany). After deparaffinization, sections were stained for collagen fibers using a Picro Sirius red stain kit (Abcam, London, UK; ab150681). The standard protocol recommended by the manufacturer was used, including using muscle samples as negative controls. The semi-quantitative analysis of total collagenous tissue maturation was carried out according to Maia-Figueiro et al. [66] with a slight modification; the experimental samples were classified in the following three grades of red intensity in comparison with the red intensity of the control sample: “0” (equal intensity—same maturation); “−” (lower intensity—less advanced maturation); “+” (higher intensity—more advanced maturation).

### 4.7. Determination of Hydroxyproline

A total of 50 mg of the sample was mixed with 1 mL of 2 M HCl and incubated at laboratory temperature for 20 min. The solution was subsequently subjected to digestion in a microwave reactor (Anton Paar GmbH, Graz, Austria) using the following conditions: power of 80; ramp up for 10 min and hold for 90 min; maximum temperature of 120 °C; maximum pressure of 25 bar. After mineralization, samples were evaporated using a Stuart P-LAB a. s. nitrogen blow-down evaporator with spiral needles (Bibby Scientific Ltd., Stone, UK). The dried sample was dissolved in 200 μL of 0.1 M HCl, vortexed, transferred to an Eppendorf tube and centrifuged at 11,200 for 10 min at 4 °C. The supernatant was removed, filtered (Whatman, Mini-Uniprep filters; GMF; 0.45 μm) and pipetted to a vial.

The pre-column derivatization of the sample was performed using FMOC reagent (9-fluorenylmethyl chloroformate; Agilent Technologies, Santa Clara, CA, USA) according to instructions of the column manufacturer. Hydroxyproline was separated by HPLC using a Zorbax Eclipse AAA column (150 mm × 4.6 mm; particle size of 3.5 μm; Agilent Technologies) with a Zorbax Eclipse AAA guard column (12.5 mm × 4.6 mm; particle size of 5 μm; Agilent Technologies) and an Agilent 1260 Infinity II liquid chromatography system (Agilent Technologies). The column was thermostatted at 40 °C, and the flow rate of the mobile phase was 2 mL/min. Mobile phase A consisted of 40 mM Na_2_HPO_4_ at pH 7.8 (5.5 g of NaH_2_PO_4_ monohydrate/1 L of H_2_O adjusted to pH 7.8 with 10 mM NaOH), and mobile phase B consisted of acetonitrile/methanol/water (45:45:10 *v*/*v*/*v*). The analyte was eluted with a linear upward gradient: 0% B for 0.0 min → 0% B for 1.9 min → 57% B for 18.1 min →100% B for 18.6 min → 100% B for 22.3 min → 0% B for 23.2 min → 0% B for 26 min. The column effluent was monitored with a fluorescence detector at excitation and emission wavelengths of 340 and 450 nm, respectively. Trans-4-hydroxy-L-proline, >99% (Sigma Aldrich), was used as an external standard, and data were evaluated according to a calibration line using calibration solutions of 0.8–40 μg/mL.

### 4.8. Evaluation of the Extent of Angiogenesis

Cryosections were rehydrated and pretreated with citrate buffer (pH = 6; 0.01 M) in a hot water bath at 98 °C for 15 min. For the inhibition of non-specific secondary antibody binding, the sections were incubated with a blocking serum (Vectastain ABC Kit; Rabbit IgG; PK-4001; Vector Laboratories, Burlingame, CA, USA) for 20 min at room temperature (RT). The sections were then incubated with the primary antibody (rabbit alpha-SMA; CD68; Abcam, London, UK; Cat. No. ab5694; diluted to 1:100) for 60 min at RT. After the application of the biotinylated secondary antibody (Vectastain) for 30 min at RT, the slices were incubated with a peroxidase-conjugated avidin–biotin complex (Vectastain) for 30 min at RT. The chromogen substrate, diaminobenzidine (Liquid DAB + Substrate Chromogen System; K3468; Agilent Dako, Santa Clara, CA, USA), was used for the visualization of positive cells. Blood vessels were recognized by the presence of smooth muscle cells expressing alpha-SMA using QuickPHOTO MICRO software, version 3.2 (Promicra, Prague, Czech Republic). The number of vessels was counted in three 333 µm × 333 μm squares localized at three different depths on the edge of the healing tissue; the total number of blood vessels was calculated as a sum of the counts from the three squares.

### 4.9. Quantification of the Gene Expression

Tissue samples were stored in RNAlater with RNA Stabilization Reagent (Qiagen, Hilden, Germany) at 4 °C according to the manufacturer’s instructions and further homogenized in a stabilization reagent (TRI Reagent RT; MRC, Cincinnati, OH, USA).

Total RNA was isolated using 4-bromoanisole and purified using NucleoSpin RNA Plus Mini Kit (Macherey Nagel, Düren, Germany), including the gDNA removal column, according to the manufacturer’s animal protocol. The RNA purity and integrity were confirmed spectrophotometrically (absorbance ratios at A260/A280 nm and A260/A230 nm) and by indication of 28S and 18S rRNA on agarose gel. mRNA was reverse-transcribed using Luna Script RT SuperMix Kit (NEB, Ipswich, MA, USA), and cDNA was stored at −20 °C for qPCR use.

Gene expression was measured in triplicate 3 μL reactions in a 384-well plate using a LightCycler 480 real-time PCR system (Roche, Basel, Switzerland) according to the manufacturer’s recommendations. The conditions for the qPCR reaction were as follows: initial denaturation at 95 °C for 15 min; then, 50 cycles with denaturation at 95 °C for 15 s; primer annealing at 58 °C for 30 s; and elongation at 72 °C for 30 s. The reaction mixture included 10 pmol of each of the primers (Generi Biotech, Hradec Kralove, Czech Republic), 5× diluted cDNA (0.5 μL) and 1.5 μL of QuantiTect SYBR Green PCR MasterMix (Qiagen, Darmstadt, Germany). A Nanodrop II liquid dispenser (IDEX Health & Science LLC, Oak Harbor, WA, USA) was used for pipetting.

The list of primers used in our qPCR analyses is presented in Table 2. TBP1 was selected as reference gene (RG) among three tested housekeeping genes due to its low variability across all samples tested, as determined using the NormFinder algorithm (NormFinder, Aarhus University Hospital, Denmark [67]). The reference gene was also used for confirmation of the qPCR correct course. Assuming a primer efficiency ≥1.9, the normalized expression of the target gene (TG) based on the threshold cycle values (Cq) [68] was calculated as 2 ^− [(CqTGsample − CqRGsample) − (CqTGcontrol − CqRGcontrol)]^ [69]. The amplicon melting temperature analysis was obtained. To detect contamination by nucleic acid, negative controls were included at each individual step of the gene expression analysis.

### 4.10. Microbiological Analysis

Samples were obtained by smearing pig skin-excision wounds (5 cm × 5 cm) using sterile flocked swabs (VWR, Radnor, PA, USA) 0, 7, 14 and 21 days after excision. Swabs were eluted in 10 mL of sterile Ringer’s solution (Neogen, Lansing, MI, USA). One milliliter of selected (based on the presumptive degree of contamination) 10-fold dilution of each sample was applied to the parallel agar plates. Counts of viable cells of total microorganisms (TMCs) were obtained for three species (*Staphylococcus aureus*, *Streptococcus pyogenes* and *Escherichia coli*) as described below.

The determination of the TMCs was performed using Plate Count Agar (PCA; Neogen, Lansing, MI, USA) plates that were incubated at 30 °C for 72 h under aerobic conditions. Selective bacteria isolations were carried out as follows: *Staphylococcus aureus* on Aureus ChromoSelect Agar Base (Sigma-Aldrich, St. Louis, MO, USA) supplemented with Egg yolk tellurite emulsion (HiMedia, Mumbai, India), incubated at 37 °C for 24–48 h under aerobic conditions; *Streptococcus pyogenes* on blood agar (Thermo Fisher Scientific, Waltham, MA, USA) supplemented with defibrinated horse blood (Thermo Fisher Scientific, Waltham, MA, USA), incubated at 37 °C for 24–48 h under aerobic conditions; *Escherichia coli* on Harlequin Chromogenic Coloform Agar (Neogen, Lansing, MI, USA), incubated at 37 °C for 24–48 h under aerobic conditions.

The presumptive colonies of *S. pyogenes* were tested for PYR activity (ITEST Plus, Hradec Kralove, Czech Republic), and confirmation was performed using MIKRO-LA-TEST Streptotest 24 biochemical tests (Erba-Lachema, Brno, Czech Republic).

The confirmations of *S. aureus* and *E. coli*, respectively, were performed using strain-specific polymerase chain reaction (PCR). The DNA extraction from purified presumptive colonies were performed with a NucleoSpin Tissue purification kit (Macherey Nagel, Denmark). For PCR identification, the following specific primers were used: for *S. aureus,* SA-1 (5′-GCGATTGATGGTGATACGGTT-3′)/SA-2 (5′-CAAGCCTTGACGAACTAAAGC-3′) [74] amplifying a 276 bp DNA fragment; for *E. coli,* Eco-1 (5′-GACCTCGGTTTAGTTCACAGA-3′)/Eco-2 (5′-CACAC-GCTGACGCTGACCA-3′) [75] amplifying a 585 bp DNA fragment. For both reactions, the PCR mixture contained 2.5 μL of 10× reaction buffer, 0.5 μL of 10 mM of dNTPs, 1.0 μL of 10 pmol/L of each primer, 1.0 μL of 1U Taq DNA polymerase and up to 25 μL PCR water (Taq PCR Master Mix Kit, Qiagen, Denmark). Amplification using these primers targeted the staphylococcal nuclease gene (nuc) and malB promoter gene (malB) of *E. coli*.

Reactions for both primer pairs were carried out with an initial denaturation at 94 °C for 2 min, 35 amplification cycles (denaturation at 94 °C for 30 s, annealing at 50 °C for 30 s and elongation at 74 °C for 35 s) and a final extension at 74 °C for 2 min (MJ Mini Cycler Bio-Rad, Hercules, CA, USA). The detection of the PCR products was performed with agarose gel electrophoresis in 1.8% agarose gels (SERVA, Slangerup, Denmark) stained with DNA Stain G (SERVA, Denmark) and visualized using a UV transilluminator (Ultra Lum, Berlin, CTUSA).

### 4.11. Statistical Evaluations

The normality of the data distribution was tested with the Kolmogorov–Smirnov test. The differences among treatments were evaluated with the one-way analysis of the variance ratio test, including Tukey’s post hoc test; differences were considered significant at the level of *p* < 0.05. The STATISTICA 12 package (StatSoft, Tulsa, OK, USA) was used for evaluations.

## 5. Conclusions

Using a porcine model to study wound healing, we found that fish oil entrapped in PLGA NPs did not significantly improve the markers of cutaneous wound healing in comparison with FO alone. On the other hand, the effect of PLGA-NP-entrapped fish oil was comparable to that of PLGA-NP-entrapped mupirocin. NP/FO treatment can, therefore, be suggested as a suitable alternative to NP/MUP in cutaneous wound treatment, with the additional advantage of decreasing the risk of bacterial resistance.

## Figures and Tables

**Figure 1 ijms-23-07663-f001:**
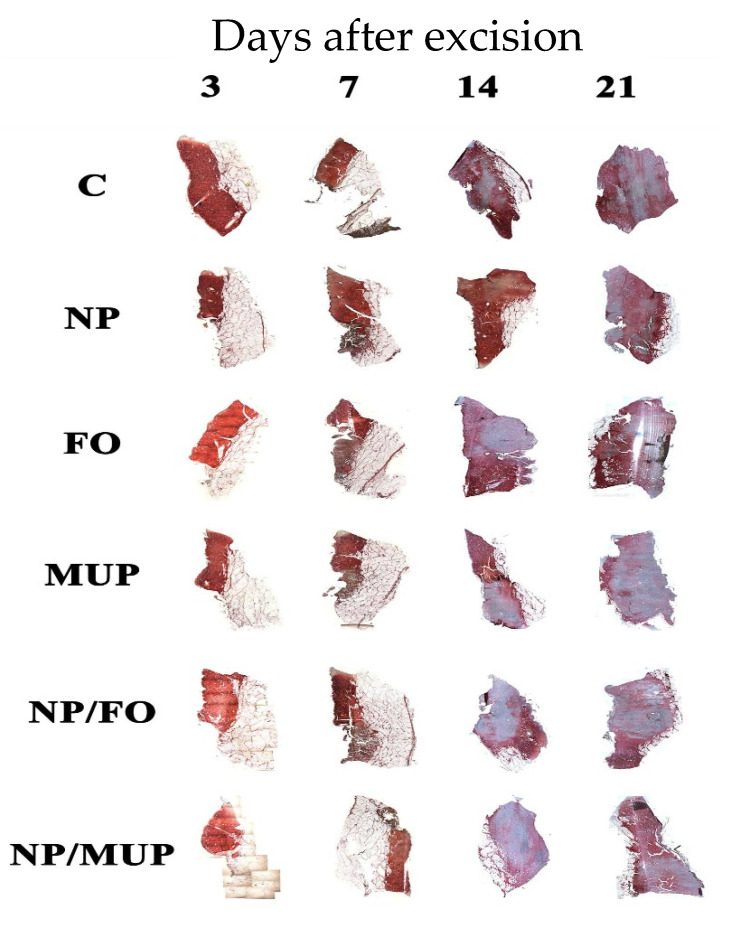
Collagenous tissue maturation (representative pig) semi-quantified using Sirius red staining. C—control (dressing alone); NP—dressing with empty PLGA-PVA nanoparticles (NPs); FO—dressing with fish oil; MUP—dressing with mupirocin; NP/FO—dressing with PLGA-PVA NPs with entrapped fish oil; NP/MUP—dressing with PLGA-PVA NPs with entrapped mupirocin. Histological images are magnified 10×.

**Figure 2 ijms-23-07663-f002:**
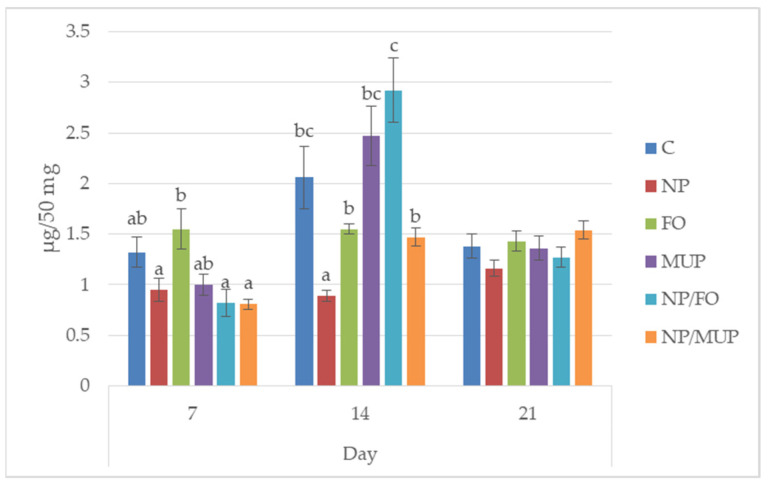
Hydroxyproline content in the wound in the course of healing (7, 14 and 21 days post-excision). C—control (dressing alone); NP—dressing with empty poly(lactic-co-glycolic) acid (PLGA)-polyvinyl alcohol (PVA) nanoparticles (NPs); FO—dressing with fish oil; MUP—dressing with mupirocin; NP/FO—dressing with PLGA-PVA NPs with entrapped fish oil; NP/MUP—dressing with PLGA-PVA NPs with entrapped mupirocin. Mean ± SEM. a–c: means with different superscripts within a given time interval differed significantly (*p* < 0.05). One-way analysis of the variance ratio test with Tukey’s post hoc test.

**Figure 3 ijms-23-07663-f003:**
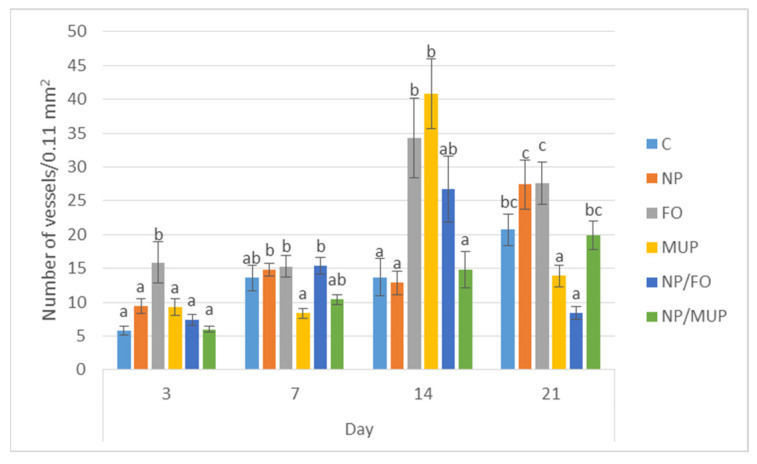
The semi-quantitative analysis of the extent of neoangiogenesis in the wound in the course of healing (3, 7, 14 and 21 days post-excision) using immunohistochemical labeling of vessels for alpha-smooth muscle actin. C—control (dressing alone); NP—dressing with empty poly(lactic-co-glycolic) acid (PLGA)-polyvinyl alcohol (PVA) nanoparticles (NPs); FO—dressing with fish oil; MUP—dressing with mupirocin; NP/FO—dressing with PLGA-PVA NPs with entrapped fish oil; NP/MUP—dressing with PLGA-PVA NPs with entrapped mupirocin. Mean ± SEM. a–c: different superscripts within a given time interval indicate a significant difference (*p* < 0.05). One-way analysis of the variance ratio test with Tukey’s post hoc test.

**Figure 4 ijms-23-07663-f004:**
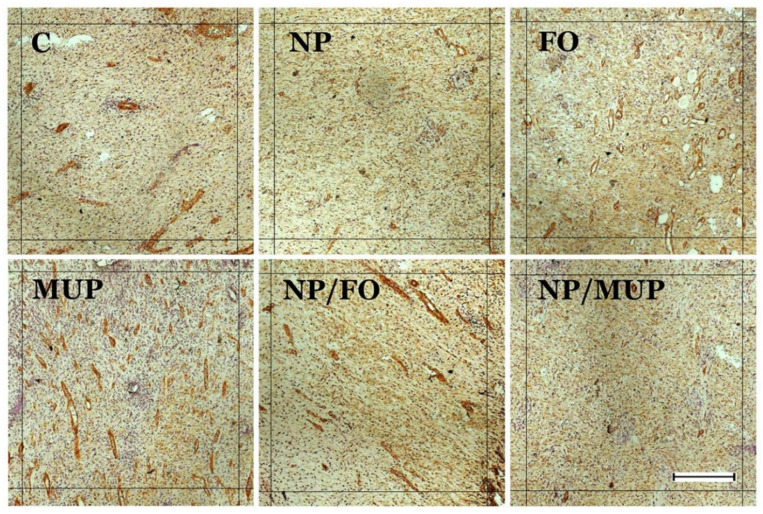
Representative preparations of the healing tissue 14 days post-excision (immunohistochemical labeling of vessels for alpha-smooth muscle actin). C—control (dressing alone); NP—dressing with empty poly(lactic-co-glycolic) acid (PLGA)-polyvinyl alcohol (PVA) nanoparticles (NPs); FO—dressing with fish oil; MUP—dressing with mupirocin; NP/FO—dressing with PLGA-PVA NPs with entrapped fish oil; NP/MUP—dressing with PLGA-PVA NPs with entrapped mupirocin; scale bar = 100 µm.

**Figure 5 ijms-23-07663-f005:**
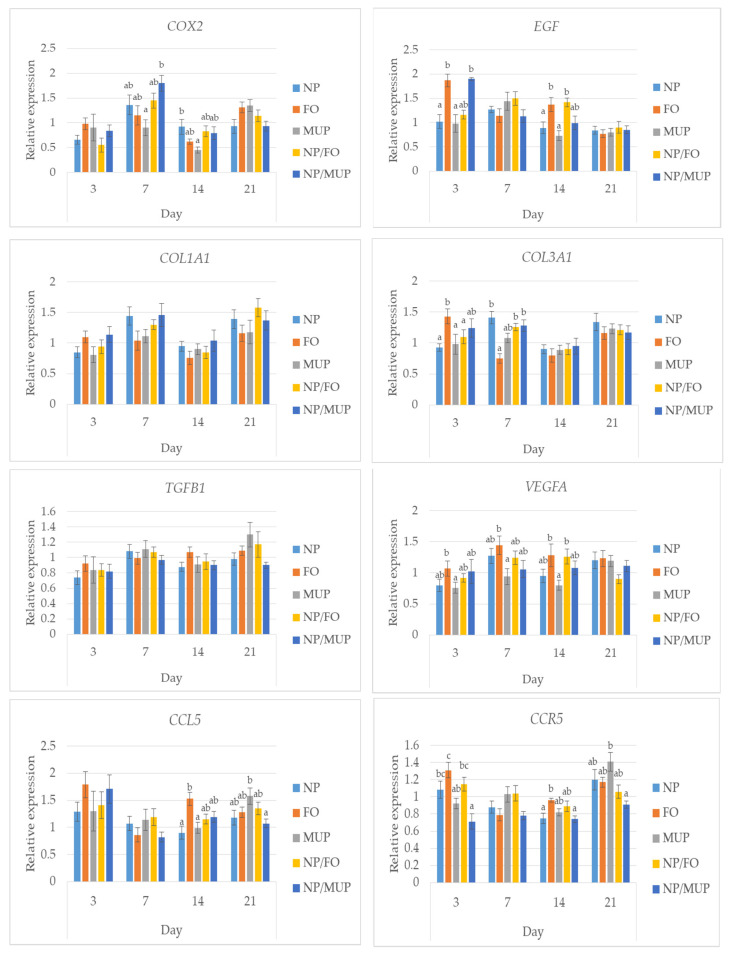
Relative expressions of the genes relevant to inflammation (prostaglandin-endoperoxide synthase 2, *COX2*), cell growth and differentiation (epidermal growth factor, *EGF*), collagen formation (alpha-1 type I collagen, *COL1A1*; alpha-1 type III collagen, *COL3A1*), cell proliferation and wound healing (transforming growth factor beta-1, *TGFB1*), angiogenesis (vascular endothelium growth factor alpha, *VEGFA*), and chemotaxis and immune cells recruitment (chemokine-C-C motif ligand 5, *CCL5*; C-C chemokine receptor type 5, *CCR5*). NP—dressing with empty poly(lactic-co-glycolic) acid (PLGA)-polyvinyl alcohol (PVA) nanoparticles (NPs); FO—dressing with fish oil; MUP—dressing with mupirocin; NP/FO—dressing with PLGA-PVA NPs with entrapped fish oil; NP/MUP—dressing with PLGA-PVA NPs with entrapped mupirocin. Mean ± SEM. a–c: means with different superscripts within a given time interval differed significantly (*p* < 0.05); One-way analysis of the variance ratio test with Tukey’s post hoc test.

**Figure 6 ijms-23-07663-f006:**
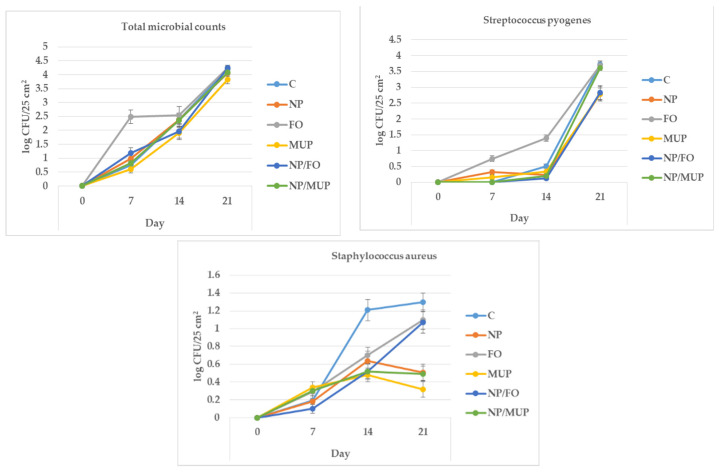
Change in microbial counts over time after excision for wounds treated with dressing alone (control, C); dressing with empty poly(lactic-co-glycolic) acid (PLGA)-polyvinyl alcohol (PVA) nanoparticles (NPs) (NP); dressing with fish oil (FO); dressing with mupirocin (MUP); dressing with PLGA-PVA NPs with entrapped fish oil (NP/FO) and dressing with PLGA-PVA NPs with entrapped mupirocin (NP/MUP).

**Figure 7 ijms-23-07663-f007:**
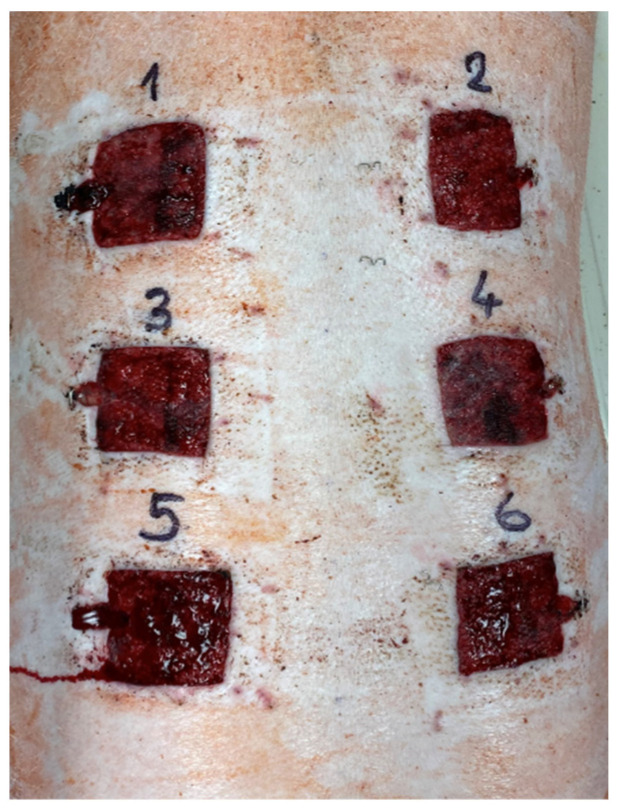
Position of the wounds on pig’s dorsum (state 7 days post-excision).

**Table 1 ijms-23-07663-t001:** Visually assessed differences from control (dressing alone) of the total collagenous tissue by the semi-quantitative analysis using the Sirius Red staining (0 same amount of collagen; – less amount of collagen).

Sample	Days after Excision			
	3	7	14	21
NP ^1^	0	–	–	–
FO ^2^	0	–	–	0
MUP ^3^	0	–	0	0
NP/FO ^4^	0	0	–	0
NP/MUP ^5^	0	–	–	0

^1^ Dressing with empty poly(lactic-co-glycolic) acid (PLGA)-polyvinyl alcohol (PVA) nanoparticles (NPs). ^2^ Dressing with fish oil. ^3^ Dressing with mupirocin. ^4^ Dressing with PLGA-PVA NPs with entrapped fish oil. ^5^ Dressing with PLGA-PVA NPs with entrapped mupirocin.

**Table 2 ijms-23-07663-t002:** List of gene-specific primers used in qPCR.

Gene	5′-Forward Primer-3′ 5′-Reverse Primer-3′	Primer Origin Amplicon Length ^1^/Eff ^2^
*COX2*	CTTAAACAGGAGCACCCGGAATCACAATCTTAATCGTTTCTCCTATCAG	designed in this study 87/2.053
*EGF*	AGCTATGCCTGCAACTGTGTTT ACCATTTCAAGTCTCTGTGCTGAC	designed in this study 67/1.945
*COL1A1*	ACGCCATCAAAGTCTTCTGCAAC TTGGGGTTCTTGCTGATGTACCA	designed in this study 103/2.051
*COL3A1*	GACGAGATGGAAACCCTGGATCAAGGAGAGCCATTTTCACCACGAT	designed in this study 89/2.040
*TGFb1*	TACGCCAAGGAGGTCACCC CAGCTCTGCCCGAGAGAGC	von der Hardt et al. [70] 156/2.007
* VEGFA *	TAGAGCGAGGCAAGAAAATCCCT CAGGAACATTTACACGTCTGCGG	designed in this study 90/2.089
*CCL5* (*RANTES*)	ACCACACCCTGCTGTTTTTC GGCGGTTCTTTCTGGTGATA	Ondrackova et al. [71] 124/2.014
*CCR5*	TGGTCAGAGGAGCTGAGACA AGAAGGGACTCGTCGTTTGA	Ondrackova et al. [70] 86/2.084
*TBP1*	AACAGTTCAGTAGTTATGAGCCAGA AGATGTTCTCAAACGCTTCG	Nygard et al. [72] 153/1.938

^1^ The size of the amplicons (in number of nucleotides) for the primer sets was derived using Primer BLAST (https://www.ncbi.nlm.nih.gov/tools/primer-blast/ (accessed on 18 January 2022)) [73] on the basis of the current nucleotide sequences available in GenBank (National Center for Biotechnology Information, Pike, Bethesda, MD, USA). The primer sets with at least one of the primers at the mRNA exon–exon junction were preferred to eliminate genomic DNA amplification. ^2^ The efficiency of the primer sets was evaluated based on the 10-fold dilution series using LightCycler 480 Software 1.5.1.62 (Roche Diagnostics GmbH, Roche Applied Science 68298, Mannheim, Germany, 2008).

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
