# Peer review of "Effect of Polymeric Nanoparticles with Entrapped Fish Oil or Mupirocin on Skin Wound Healing Using a Porcine Model"

_ijms, 2022, doi:10.3390/ijms23147663_

Round 1

Reviewer 1 Report

This study used collagen scaffold (control, C); empty poly(lactic-co-glycolic) acid PLGA (NP), fish oil (FO), mupirocin (MUP), PLGA NPs with entrapped FO (NP/FO), and PLGA NPs with MUP (NP/MUP) to treat pig skin wounds. Collagen content, angiogenesis, gene expression of COX2, EGF, COL1A1, 27 COL1A3, TGFB1, VEGFA, CCL5 and CCR5, and bacterial colonization were evaluated. Here are my major and minor comments:

Major comments,

1: The paper was poorly written. It is very hard to understand/follow the results described. The conclusion is not consistent with the data shown.

2: No wound closure data

Minor comments:

1: Table 1: give details of the method for vision assessment of the collagen content.

2: Fig 1: no detailed description of the result of staining. Staining in Day 14 and 21 images seems like Masson’s Trichrome staining.

3: Fig. 1 and 4 need scale bars.

4: All figures: lack of the definitions of labels of a, b, c etc for significance. The description of the results was very confusing.

5: Line 531: what kind of collagen was used?

6: Line 543-545: the authors described:  NP/FO 50mg, NP/MUP 10mg, how much of each component was in NP/FO or NP/MUP?

7: Line 646: How was primer efficiency determined?

Author Response

List of changes – answers to the reviewers' comments

Rev. 1

Major comments

1: The paper was poorly written. It is very hard to understand/follow the results described.

Answer: The paper was thoroughly revised. The text was also corrected by a native speaker. The results section was simplified, with emphasis on comparison between entrapped fish oil and fish oil alone, and entrapped fish oil with entrapped mupirocin, respectively.

The conclusion is not consistent with the data shown.

Answer: Conclusions were reformulated based on the results of the present study.

2: No wound closure data.

Answer: The objective of the present study was not to monitor the excisions until the wound closure, but formation of the granulation tissue and a smooth transition from the inflammatory phase to the proliferative phase, which, from the clinical viewpoint, was successful. The total wound closure of the full-thickness excisions (used in the present experiment) is a question of many weeks or even months and it cannot been expected within the three weeks of this study.

Minor comments:

1: Table 1: give details of the method for vision assessment of the collagen content.

Answer: Details regarding classification of the intensity of the Sirius red staining were included in Table 1.

2: Fig 1: no detailed description of the result of staining. Staining in Day 14 and 21 images seems like Masson’s Trichrome staining.

Answer: We added the following text: „Collagenous tissue … maturation was assessed by the red intensity. In Figure 1, collagen type III and collagen type I is characterized by the light red color and the dark red color, respectively. It is evident that most of the collagen type I was detected on day 14 and 21.

3: Fig. 1 and 4 need scale bars.

Answer: We added scale bar to Figure 4. However, Figure 1 is composed of 24 partial histological images and the scale bar would have to be inserted only into one of them. But in that case the scale bar will not be clearly visible. So we only added to Figure 1 a note regarding magnification of the whole histological image: "Magnification of histological image 10x".

4: All figures: lack of the definitions of labels of a, b, c etc for significance. The description of the results was very confusing.

Answer: The labels, including a significance level, are defined in the revised MS in all figures where the results are expressed in the form of the bar chart (Figures 2,3,5):  “a – c: means with different superscripts within a given time interval differ significantly (p < 0.05)”. The method of statistical evaluation is also mentioned in these figures.

And a statistical analysis section was included in the manuscript.

5: Line 531: what kind of collagen was used?

Answer: The word “bovine” was added into the text.

6: Line 543-545: the authors described:  NP/FO 50mg, NP/MUP 10mg, how much of each component was in NP/FO or NP/MUP?

Answer: We added the data regarding respective amounts of FO and MUP per one gram of NPs (part 4.3.) and amounts per actual content of NPs in the dressing (part 4.4.), respectively.

7: Line 646: How was primer efficiency determined?

Answer: Reference to the Roche Diagnostics software used for determination of the primer sets was added to Table 2.

Reviewer 2 Report

Abstract

Lines 33-34 mupirocin does not possesses healing activity itself. Is antibiotic and therefore direct comparison between healing activities of mupirocin and fish oil ca vot be decribed. The phrase has to be rewritten

It is a very well organised  work and the authors describe exactly the “controversial “ results

I think that phrases that show direct comparison between healing effects of mupirocin and fish oil have to be omitted, although angiogenic effects of mupirocin have been referred.

Author Response

List of changes – answers to the reviewers' comments

Rev. 2

Lines 33-34 mupirocin does not possesses healing activity itself. Is antibiotic and therefore direct comparison between healing activities of mupirocin and fish oil ca vot be decribed. The phrase has to be rewritten

Answer: We replaced the term “healing” by the term “treatment”.

It is a very well organised work and the authors describe exactly the “controversial “ results

I think that phrases that show direct comparison between healing effects of mupirocin and fish oil have to be omitted, although angiogenic effects of mupirocin have been referred.

Answer: We omitted direct comparison between effects of FO alone and MUP alone in the whole text (both in the results and in the discussion). We put emphasis only on the differences between effects of entrapped fish oil and fish oil alone, and between entrapped fish oil and entrapped mupirocin, respectively.

Reviewer 3 Report

The manuscript needs extensive revision because many methods and results are not clearly described and explained.

Authors need to provide clear evidence and proof of their results. Almost all methods sections need to be revised.

A few comments in particular:

Nanoparticles should be better characterized and described in the methods section.

Results and characterization of nanoparticles should be reported in the results section with data on Chemical and physical characterization studies, in terms of size, homogeneity, zeta potential,, cloud point, encapsulation efficiency, load capacity, in vitro release, and storage stability, on empty PLGA 24 NPs (NP), FO, mupirocin (MUP), PLGA NPs with entrapped FO (NP/FO), and PLGA NPs with entrapped MUP (NP/MUP).

Figure 1 is of poor quality and therefore difficult to evaluate.

Line 587 "Fifty μg of sample was hydrolyzed with 500 μl of 6 M HCl", the Authors need to explain how they managed to fifty μg of sample.

Table 2 needs to be presented more clearly and comprehensively

Line 561 microbiological analysis must be described in the methods section correctly and completely

Line 611 "citrate buffer (pH = 6)" concentration should be indicated.

Author Response

List of changes – answers to the reviewers' comments

Rev. 3

The manuscript needs extensive revision because many methods and results are not clearly described and explained.

Answer: We revised the MS extensively, including the M&M section. The text of the results was simplified, emphasis was put mainly on comparison of entrapped fish oil with fish oil alone, and entrapped fish oil with entrapped mupirocin, respectively.

Authors need to provide clear evidence and proof of their results. Almost all methods sections need to be revised.

Answer: The Methods sections were revised and corrected.

A few comments in particular:

Nanoparticles should be better characterized and described in the methods section.

Results and characterization of nanoparticles should be reported in the results section with data on Chemical and physical characterization studies, in terms of size, homogeneity, zeta potential,, cloud point, encapsulation efficiency, load capacity, in vitro release, and storage stability, on empty PLGA 24 NPs (NP), FO, mupirocin (MUP), PLGA NPs with entrapped FO (NP/FO), and PLGA NPs with entrapped MUP (NP/MUP).

Answer: Thank you for your suggestion. Characteristics of the NP/FO have already been described and published in the paper Komprda et al., Poly(lactic-co-glycolic) acid nanoparticles as delivery system of fish oil for wound healing, Acta Veterinaria Brno 2022, 91(3), in print. As far as NP/MUP nanoparticles are concerned, their characteristics are described in paper Popelkova et al., Poly(lactic-co-glycolic) acid nanoparticles as feasible delivery system of antimicrobial mupirocin for possible wound healing, submitted to RSC Advances. We present here only the most important NPs characteristics, in the M&M section, part 4.3 “Production of nanoparticles” rather than in the Results section.

Figure 1 is of poor quality and therefore difficult to evaluate.

Answer: Figure 1 has been enlarged, which improved its quality. We also added in the text a note concerning evaluation of Figure 1: „Collagenous tissue … maturation was assessed by the red intensity. In Figure 1, collagen type III and collagen type I is characterized by the light red color and the dark red color, respectively. It is evident that most of the collagen type I was detected on day 14 and 21.“

Line 587 "Fifty μg of sample was hydrolyzed with 500 μl of 6 M HCl", the Authors need to explain how they managed to fifty μg of sample.

Answer: We apologize for a mistake in the order of magnitude. We corrected the first part of the 4.7. section.

Table 2 needs to be presented more clearly and comprehensively

Answer: Because Table 2 seemed not to be clear, we omitted it altogether and we mention the relevant correlations only in the text.

Line 561 microbiological analysis must be described in the methods section correctly and completely

Answer: Microbiological analysis is completely described in the M&M section.

Line 611 "citrate buffer (pH = 6)" concentration should be indicated.

Answer: The concentration of the citrate buffer was added (1st line of the part 4.8.: 0.01 M).

Round 2

Reviewer 1 Report

The authors made a great effort to revise the paper. I have no more comments. 

Reviewer 3 Report

The manuscript has been improved.

Because of the many corrections, the format of this revised manuscript is difficult to read, it would have been helpful to have a final revised format that is easier to read as well